# Multifaceted Effect of *Rubus Occidentalis* on Hyperglycemia and Hypercholesterolemia in Mice with Diet-Induced Metabolic Diseases

**DOI:** 10.3390/nu10121846

**Published:** 2018-12-01

**Authors:** Jiyeon Kim, Jinho An, Heetae Lee, Kyungjae Kim, Su Jung Lee, Hye Ran Choi, Ji-Wung Kwon, Tae-Bum Lee, Youngcheon Song, Hyunseok Kong

**Affiliations:** 1College of Pharmacy, Sahmyook University, Seoul 01795, Korea; shelly7285@naver.com (J.K.); romang1230@naver.com (J.A.); hite486@gmail.com (H.L.); kimkj@syu.ac.kr (K.K.); 2Berry & Biofood Research Institute, Jeonbuk 585-943, Korea; 2-su@hanmail.net (S.J.L.); 111326@hanmail.net (H.R.C.); kjwung@hanmail.net (J.-W.K.); 3Institute of natural cosmetic industry for namwon, Jeollabuk-do 55801, Korea; tblee01@gmail.com; 4College of Animal Biotechnology & Resource, Sahmyook University, Seoul 01795, Korea

**Keywords:** *Rubus occidentalis*, black raspberry, anti-diabetic effect, cholesterol-lowering effect, diet-induced metabolic diseases model

## Abstract

Metabolic syndrome is characterized by a combination of several metabolic disorders, including obesity, hyperglycemia, and hyperlipidemia. A simultaneous occurrence is one of the most crucial features of metabolic syndrome; therefore, we selected an animal model in which this would be reflected. We fed C57BL/6N mice a high-fat diet for 23 weeks to develop metabolic syndrome and examined the efficacy of *Rubus occidentalis* (RO) for hyperglycemia and hypercholesterolemia. Oral administration of RO for 16 weeks improved hyperglycemia as indicated by significantly decreased fasting glucose levels and a glucose tolerance test. Improvements were also observed in hypercholesterolemia, in which significant decreases in serum total cholesterol, non-high-density lipoprotein (non-HDL) cholesterol, apolipoprotein A-1, and apolipoprotein B levels were observed. The time comparison of major biomarkers, observed at the initiation and termination of the experimental period, consistently supported the beneficial effects of RO on each metabolic phenotype. In addition, RO treatment attenuated the excessive fat accumulation in hepatic and adipose tissue by decreasing the size and number of lipid droplets. These results suggested that RO simultaneously exerted antihyperglycemic and antihyperlipidemic effects in mice with diet-induced metabolic syndrome.

## 1. Introduction

One of the most prevalent diseases in modern society is metabolic syndrome, which is closely related to obesity. Metabolic syndrome is a serious threat to public health because it is closely related to the modern lifestyle and issues such as lack of exercise, a poor-quality diet, and aging. Metabolic diseases are generally induced by obesity, which increases fatty acid content in the body. As a result, triglycerides are elevated and diabetes is caused by insulin resistance [1]. Metabolic syndrome is defined as a cluster of two or three diseases; for example, concomitant hyperglycemia and hyperlipidemia caused by obesity [2]. When disease symptoms develop in a complex manner, the risk of serious complications, such as stroke or diabetes, is two to five times higher than when the disease symptoms occur alone [3]. Therefore, the United States and most European countries have considered metabolic syndrome a grave public health problem for decades and, relatively recently, South Korea started to study preventative care for metabolic syndrome [4,5].

*Rubus* species, including raspberries, blackberries, and dewberries, have gained attention as novel therapeutic agents for metabolic syndrome because of their anti-inflammatory effects [6]. The antioxidant and organic acid contents have been shown to contribute to its beneficial effects [7]. *Rubus occidentalis* (RO), commonly called the “black raspberry”, is one of the most popular fruits used as a flavorant and additive. It has been used for nourishment and as traditional remedies, especially in the immature form, in Asian countries [6,8]. Numerous studies have reported that the active compounds present in this fruit include various antioxidants, such as flavonoids and organic acids, and have investigated its significant antitumor, anti-inflammatory, and lipid metabolism effects [9,10,11]. However, for commercial use, the quality of RO needs to be enhanced and this process has been achieved by using the black raspberry crop harvested from Gochang, Korea, the largest producer of black raspberry in Korea. Through a series of studies, various factors such as the sampling period or extraction conditions were optimized to produce fruits with a high content of bioactive principles [12,13]. In fact, the effects of *Rubus coreanus* on individual symptoms, such as its antiobesity or cholesterol-lowering effects, have previously been identified [11,14,15], while other species, especially RO, are rarely studied regarding metabolic syndrome. 

As a follow-up study on RO, in which the extract condition for the highest bioactivities established was chemically analyzed [16], we investigated the simultaneous effects of RO on metabolic phenotypes. Given that metabolic dysfunctions are related, a more comprehensive approach is required for more intensive studies and is considered clinically crucial. Therefore, we designed this study to explore the multifaceted effect of RO on metabolic syndrome. 

The diet-induced obesity (DIO) mouse is an animal model widely used for studies on metabolic syndrome because it mirrors the progression of metabolic syndrome in humans by being naturally induced with a high-fat diet and aging. However, not all obese mice develop hyperlipidemia or hyperglycemia, which means that the DIO model is not perfectly applicable to our study. Therefore, we employed a selection process before commencing the study and evaluated only animals that developed hyperlipidemia and hyperglycemia along with obesity. These animals were named diet-induced metabolic disease (DIMD) mice to distinguish them from existing obesity models. This study aimed to confirm the multifaceted effect of RO on the major metabolic phenotypes using DIMD mice.

## 2. Materials and Methods

### 2.1. Aqueous Extract of RO

RO were dried at 60 °C for 36 h and then powered for storage. RO powder was sampled and mixed with double-distilled water (DDW). The mixture was distilled twice at 80 °C for 2 h. The aqueous-extracted RO used for this study was extracted based on the standard method and was chemically analyzed as in a previous study. The RO was provided by Berry and Biofood Research Institute (Gochang, Korea) [17] Lyophilized RO was dissolved in 0.5% carboxymethylcellulose solution for homogenization and stabilization to the appropriate concentrations (125 and 250 mg/(kg·day)), which were determined using an in vitro experiment (Appendix A) following the methods of previous studies [12,17].

### 2.2. Animals and the Experimental Protocol

Male C57BL/6N mice were purchased from Samtako Co. Ltd. (Osan, Korea) and acclimated to the laboratory conditions for one week. The animals were housed in individual cages with free access to water and food in a temperature- and humidity-controlled animal facility under a 12-h light-dark cycle at 22 ± 2 °C and 55% ± 5% humidity. Mice were treated in accordance with ethical guidelines issued by the Institutional Animal Care and Use Committee (IACUC) of Sahmyook University for the care and use of laboratory animals (permission number 2015001). Egloos-type animal enrichments were distributed in each mouse cage to promote a pleasant residential environment and lessen stress. Mice were fed either a 45% kcal high-fat (lard) diet (HFD; FeedLab Inc., Guri, Korea) or a regular diet (RD; Lab rodent chow, #38057; Purina Korea Inc., Seoul, Korea).

At the 20th week of HFD feeding, the aged mice that developed a high blood glucose level (>150 mg/dL) by diabetes were first selected for further study [18]. Afterwards, blood samples were obtained by retro-orbital bleeding and analyzed using a biochemical analyzer (AU480, Beckman Coulter, Brea, CA, USA) for lipid profiling. Finally, based on a series of serum analyses, the mice that exhibited complex symptoms of diabetes, hyperlipidemia, and obesity were selected. The average values for the symptoms of blood glucose, total cholesterol, and body weight levels were 195.1 ± 32.8 mg/dL, 303.0 ± 42.5 mg/dL, and 45.0 ± 3.2 g, respectively, all of which were well above those of RD fed mice (59.4 ± 15.0 mg/dL, 124.0 ± 18.6 mg/dL, and 31.8 ± 1.5 g, respectively). All blood samples were obtained after fasting overnight.

The selected mice were divided into six groups of 10 animals per group. Ten RD-fed mice and 10 HFD-fed mice served as normal controls and disease controls, respectively. Each group was administered a vehicle once daily for 16 weeks. The rest of the DIMD mice were allocated into four groups. Each group was administered one of the following treatments once daily for 16 weeks: 125 mg/kg of RO (quantity for humans, the dose of RO; 625 mg/60 kg); 250 mg/kg of RO (quantity for humans, the dose of RO; 625 mg/60 kg); 10 mg/kg of atorvastatin (Atorva, Lipitor^®^ tablets 20 mg, Pfizer Inc., Seoul, Korea); or 250 mg/kg of metformin (Met, Diabex^®^ tab 500 mg, Daewoong Pharm. Co., Ltd., Gyeonggi-do, Korea). Atorvastatin is a “high-intensity statin”, an HMG CO-A reductase inhibitor that reduces low-density lipoprotein cholesterol (LDL-C) by more than 50% and is currently one of the most commonly used drugs in clinical practice [19]. Metformin is an antidiabetic agent that has been clinically used for decades in the treatment of type 2 diabetes mellitus and has proven effective in improving hyperglycemia [20]. For this reason, mice were treated with hypercholesterolemia and hyperglycemia, respectively, as control drugs. The amount of food consumed by mice in each treatment was measured during the experimental period, and there was no changing in the intake.

### 2.3. Blood Glucose and Intraperitoneal Glucose Tolerance Testing (IPGTT)

Throughout the experimental period, blood glucose levels of the mice were evaluated every second week. Feed was removed overnight, after which blood samples were taken from the tail and measured using Accu-chek^®^ Performa (Roche Diagnostics, Risch-Rotkreuz, Switzerland). IPGTT was performed one week before mice were sacrificed. Mice that had been fasted overnight were intraperitoneally injected with glucose solution (2 g/kg of body weight, dissolved in phosphate-buffered saline (PBS)). Blood samples were obtained from the tail vein at 0, 0.5, 1, 1.5, and 2 h after glucose administration to determine the blood glucose level at each time point.

### 2.4. Biochemical Indices in Serum and Hepatic Tissue

Serum and homogenized tissue solutions were prepared to determine lipid levels as follows. At the end of the experimental period, mice were sacrificed under ether anesthesia and blood was collected via cardiac puncture. The blood samples were centrifuged at 10,000 rpm for 5 min to isolate serum. The liver tissue and fat pad were removed and immediately frozen with liquid nitrogen before prolonged storage at −70 °C. Fifty milligrams of frozen liver tissue were then homogenized in 1 mL of PBS and centrifuged for 10 min at 4 °C and 3000 rpm. Various lipid contents (total cholesterol, triglyceride (TG), and lipoproteins) were measured in the samples using a biochemical analyzer (AU480, Beckman Coulter, Brea, CA, USA).

### 2.5. Histological Examinations

Adipose and hepatic tissue were isolated from each group of mice and fixed in 10% neutral formalin. Each tissue sample was embedded in paraffin, cut into thin sections, and mounted on glass slides for hematoxylin and eosin (H&E) staining. Sectioned tissue samples were observed with a microscope (Olympus, Tokyo, Japan) and micrographs were taken at ×200 magnification.

### 2.6. Statistical Analysis

All data are presented as mean ± standard error of mean (SEM). The differences between the control and treated groups were analyzed using one-way analysis of variance (ANOVA) followed by Dunnett’s multiple comparison test for each parameter of interest. A result of *p* < 0.05 was considered statistically significant.

## 3. Results

### 3.1. Effects of Aqueous-Extracted RO on Glucose Homeostasis

There were no significant differences in body weight between nontreated and RO-treated DIMD mice (Appendix A). RO treatment, irrespective of dose, had a significant restorative effect on fasting glucose after the 12th week of oral administration (Figure 1A). Blood glucose levels were significantly reduced at the last week of the study (Figure 1B). In addition, the higher RO dose significantly reduced the exacerbation of glucose level by the intraperitoneal glucose load within 1 h, a trend also observed in the metformin-treated mice (Figure 1C).

### 3.2. Effects of Aqueous-Extracted RO on Hyperlipidemia

As crucial indexes of hyperlipidemia, TG and total cholesterol levels were measured for each group of mice at the end of the experimental period (Figure 2). Average total cholesterol level was two to three times higher in HFD-fed mice than in RD fed mice. Total cholesterol was significantly reduced by both doses of RO (*p* > 0.05) and both drugs (*p* > 0.001) (Figure 2A). Because serum triglyceride levels are easily influenced by external variables such as fasting time or stress, but were not significantly different among groups in this study (Figure 2B), we also analyzed TG storage in hepatic tissue. As shown in Figure 2C, TG concentration in the liver decreased substantially in RO- and drug-treated mice. This decrease was consistently observed in photographs of H&E stained tissue. 

### 3.3. Effects of Aqueous-Extracted RO on Hypercholesterolemia

To explore the detailed changes in lipid profile, various lipoproteins in the serum were analyzed at the end of the experimental period (Figure 3). Although RO treatments considerably reduced the total cholesterol level in serum, high-density lipoprotein cholesterol (HDL-C) and LDL-C levels were not significantly altered (Figure 3A,B). However, apolipoprotein (Apo) indexes were significantly improved by RO treatment (Figure 3C,D). Furthermore, as a more human-applicable index of LDL-C, non-HDL cholesterol values were significantly decreased by RO and drug treatment, with the exception of the low dose of RO (Figure 3E). ApoA-1 and ApoB represent HDL-C and LDL-C, respectively, as more clinically practical indexes for disease risk.

### 3.4. Effects of Aqueous-Extracted RO on Changes in Major Biomarkers

The changes in some of the representative biomarkers during the experimental period are presented in Figure 4. As the mice aged, the fasting glucose concentration increased over 20% from baseline among the control mice but not among the treated mice. RO treatment reduced the increasing tendency in a dose-dependent manner (moderate decrease of 9.3% and 14.4%) (Figure 4A). In contrast, the average total cholesterol level of control mice naturally decreased over 16 weeks and this decreasing tendency was accelerated by oral administration of RO, atorvastatin, and metformin. In particular, RO treatment reduced total cholesterol level by more than an additional 10% (Figure 4B). Apo levels were also compared (Figure 4C,D) and exhibited different patterns of change over time. RO treatment consistently normalized each parameter by accelerating or reducing the aging tendency.

### 3.5. Effects of Aqueous-Extracted RO on Lipid Storage in Fat Pad and Hepatic Tissue

Lipid accumulation was observed in the stained tissue samples (Figure 5). The average size of adipocytes in the perirenal fat pad had increased in HFD-fed mice and decreased in mice treated with RO. However, RO treatment did not decrease adipocyte size in a dose dependent manner or as significantly as did metformin treatment, which drastically decreased lipid droplets (Figure 5A). In liver tissue, RO treatment distinctly prevented excessive fat storage (Figure 5B(c,d)). In addition, we confirmed that oral administration of RO consistently regulated lipogenesis in the liver by evaluating mRNA expression of related genes (Appendix A).

## 4. Discussion

The most characteristic trait of metabolic syndrome is the simultaneous onset of more than one pathogenic symptom, such as obesity, hyperglycemia, and hyperlipidemia [3]. Therefore, it is crucial to prepare an animal model that reflects such traits of the disease. Although RO has been recently targeted as a therapeutic agent for metabolic syndrome because of its high polyphenol and organic acid content [11,14,15], no animal models that reflect the practical traits of metabolic syndrome have been introduced to analyze the complex efficacy of RO in depth. Therefore, we studied the indicators of obesity, diabetes, and hyperlipidemia in an independent manner while those symptoms were simultaneously expressed in the same host, gaining more practical insights to the single or multifaceted efficacy of RO in metabolic syndrome. For this purpose, we developed DIMD mice with obesity, diabetes, and hyperlipidemia and used metformin and atorvastatin as control drugs.

Some notable results were obtained regarding the therapeutic effect of RO on diabetes. As shown in the results, the indicators related to glucose homeostasis (fasting glucose and IPGTT) suggested that RO significantly improved the dysfunction of glucose metabolism. This study represents the first time the antidiabetic effects of RO were investigated in metabolic syndrome. 

In addition, the oral administration of RO restored hyperlipidemia, which was reflected by a significant decrease in total cholesterol level. Several other indicators were also evaluated. There were no significant differences in TG levels between the groups. However, mouse TG is a fluctuating and changeable biomarker that frequently becomes an issue in DIO studies. Previous studies reported [12,17] that there was no significant difference in TG levels between RD and HFD groups. Under fasting conditions, TG is a readily available energy source and is present in increased levels in the blood stream, which suggests that the TG parameter could be inaccurate depending on external factors. Therefore, we measured TG storage levels in the liver and observed a decreasing tendency in the RO-treated groups compared to the control groups. 

As shown in the results, the improvement of hypercholesterolemia was primarily derived from the reduction of HDL-C, which constitutes an absolute majority of total cholesterol [21]. Considering the general consensus that HDL-C plays a positive role in hypercholesterolemia by offsetting the detrimental activity of LDL-C, it is a controversial result that RO significantly decreased HDL-C levels. However, it is important to consider the species difference in cholesterol metabolism and question whether the clinical application is reasonable. A major difference in cholesterol metabolism between mice and humans is the absence of cholesteryl ester transport protein (CETP) in mice. CETP promotes the exchange of cholesteryl ester and TG between lipoproteins. HDL-C is approximately five times higher than LDL-C in mice, whereas LDL-C is normally two to three times higher than HDL-C in humans [19]. This indicates that CETP-deficient rodents display a completely different pattern of dyslipidemia than humans and that the improvement pattern would also differ. For example, HDL-C barely influenced atherosclerosis in alcohol-treated mice, whereas in humans, HDL-C played a key role in ameliorating atherosclerosis [22]. According to Marchesi et al. [23], in human ApoA-1 transgenic mice lacking CETP, ApoA-1 plasma levels, which are directly related to HDL-C levels, did not increase with the administration of rosuvastatin, whereas rosuvastatin caused a moderate increase in plasma HDL-C concentration in humans. This suggested that the effect of statins on HDL-C levels in humans was primarily attributed to reduced CETP activity and that the therapeutic potential of rosuvastatin in clinical treatment should not be overlooked, despite its conflicting effect in mice. 

The results from our study, in which decreased HDL-C levels were observed in Atorva- and RO-treated DIMD mice, should therefore be carefully interpreted. The HDL-C-lowering effect of RO should not be considered to conflict with the amelioration of hypercholesterolemia. However, we need further studies on the relationship between HDL-C reduction by RO and CETP.

Furthermore, considering that elevated HDL-C levels caused by HFD-C were consistently observed in humans and mice [24], it is reasonable to regard this as a subsequent effect of excessive uptake of external cholesterol and not to focus on the differing roles of HDL-C. For this reason, RO treatment was evaluated in regards to how much it could normalize disease symptoms and total cholesterol was considered to be more significant. In addition, the non-HDL cholesterol level was calculated as an alternative index.

We also attempted to study the efficacy of RO in depth using Apo indicators. ApoA-1 and ApoB comprise major portions of HDL-C and LDL-C, respectively, and are responsible for the activity of each type of cholesterol, reflecting the actual and clinical disease risk [25,26]. The efficacy of RO was more significant with regard to Apo indicators than HDL-C and LDL-C, which enabled us to confirm that oral treatment of RO improved hyperlipidemia in DIMD mice in a practical manner.

Although no dramatic changes in body weight were observed over the experimental period, morphologic changes in adipocytes were observed in the stained tissue samples of RO-treated mice. In addition, the levels of lipid synthesis genes, including sterol regulatory element binding protein 1 (SREBP1), stearoyl-CoA desaturase 1 (SCD-1), and acetyl CoA carboxylase (ACC), were regulated in the liver by RO treatment. These apparent molecular-level changes suggested a significant effect of RO, even though they did not lead to phenotypic weight changes.

In conventional animal studies, it is common to confirm the target efficacy by comparing several biomarkers among groups at the end of the experimental period. This method, however, is limited because it cannot be employed to compare the biomarkers of individuals before and after experimental treatment. In this study, we have overcome the limitation by comparing biomarkers analyzed before RO administration to those obtained after termination of administration. Using the time-comparison data, we highlighted the potential of RO treatment as described above in the results section.

## 5. Conclusions

The original intention of this study was to confirm the novel efficacy of RO in metabolic syndrome. As such, we have verified the multifaceted effect of RO using DIMD mice. Oral administration of RO exhibited significant antidiabetic and cholesterol-lowering effects in a simultaneous manner. This was the first time that the therapeutic potential of RO was evaluated in terms of disease simultaneousness and complexity, which is regarded as more clinically valuable. 

## Figures and Tables

**Figure 1 nutrients-10-01846-f001:**
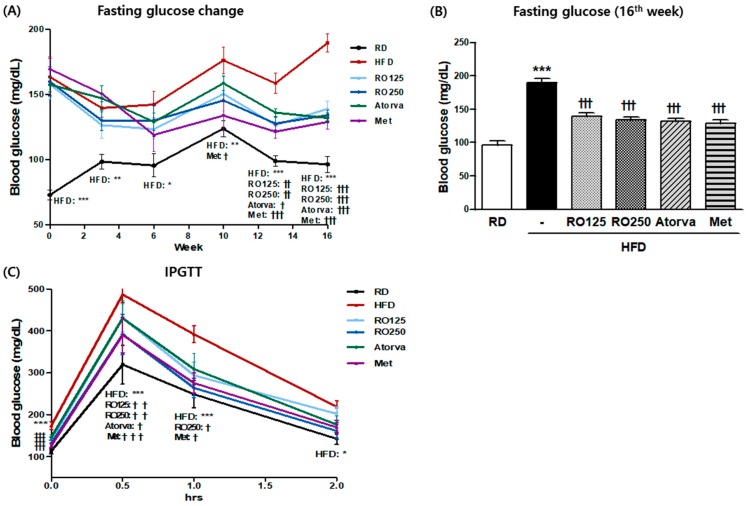
Diet-induced metabolic disease (DIMD) mice were fed an HFD and treated with *Rubus occidentalis* (RO) or drugs (atorvastatin or metformin) for 16 weeks. The changes in the fasting blood glucose (**A**) of the animals were observed every two weeks and the values from the final week are shown in (**B**). Mice glucose tolerance was analyzed by intraperitoneal glucose tolerance testing (IPGTT) (**C**). RD: regular diet, HFD: high-fat diet, HFD + RO 125: HFD-fed mice treated with a low dose (125 mg/(kg·day)) of aqueous extract of RO, HFD + RO 250: HFD-fed mice treated with a high dose (250 mg/(kg·day)) of RO. HFD + Atorva: HFD-fed mice treated with atorvastatin (10 mg/(kg·day)). HFD + Met: HFD-fed mice treated with metformin (250 mg/(kg·day)). Values are presented as mean ± standard error of mean (SEM) (*n* = 7, each group). * *p* < 0.05, ** *p* < 0.01, and *** *p* < 0.001 compared with RD mice. ^†^
*p* < 0.05, ^† †^
*p* < 0.01, and ^† † †^
*p* < 0.001 compared with HFD mice.

**Figure 2 nutrients-10-01846-f002:**
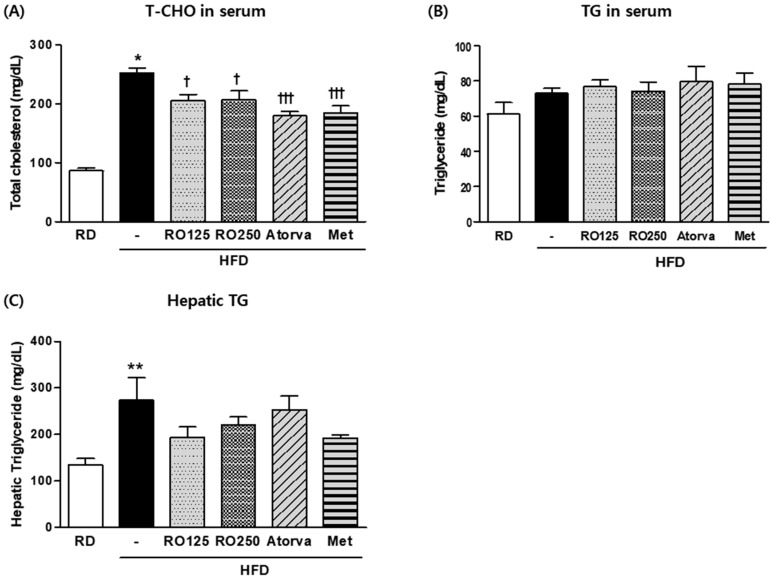
DIMD mice were fed an HFD with RO or drugs (atorvastatin or metformin) for 16 weeks. Serum from each mouse and the supernatant of their homogenized hepatic tissue solutions were prepared. Total cholesterol (T-CHO) (**A**) and triglyceride (TG) (**B**) levels in the serum as well as TG levels in the liver (**C**) were measured using a biochemistry analyzer. Mice were grouped as indicated in Figure 1. Values are presented as means ± SEM (*n* = 7, each group). * *p* < 0.05, ** *p* < 0.01, and *** *p* < 0.001 compared with RD mice. ^†^
*p* < 0.05, ^† †^
*p* < 0.01, and ^† † †^
*p* < 0.001 compared with HFD mice.

**Figure 3 nutrients-10-01846-f003:**
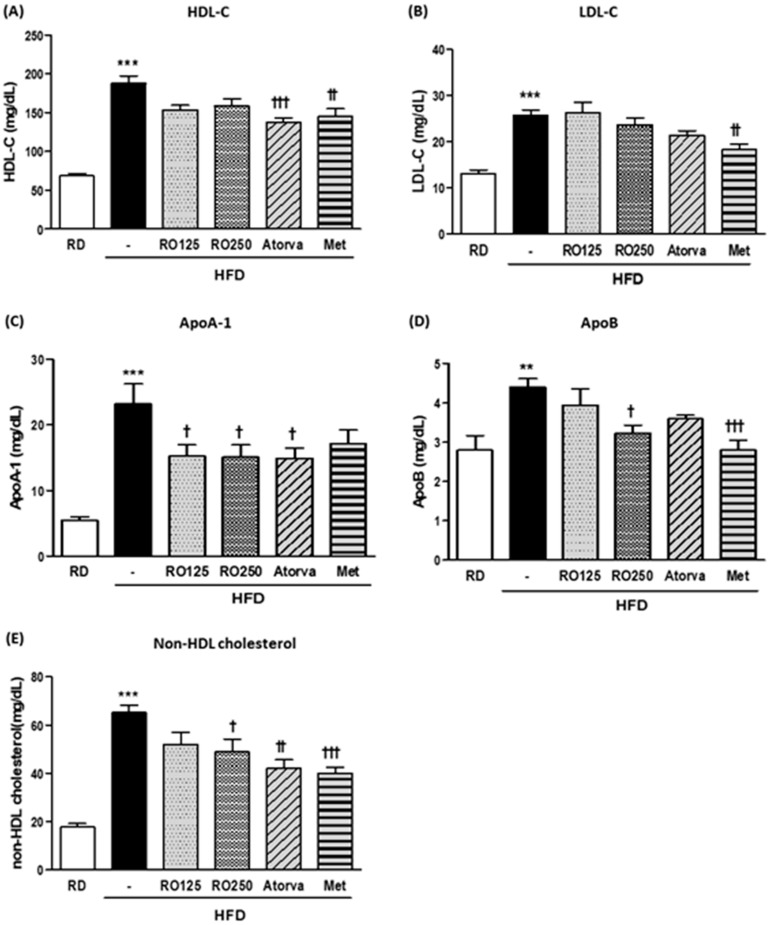
DIMD mice were fed an HFD with RO or drugs (atorvastatin or metformin) for 16 weeks. Serum from each mouse were prepared. The levels of serum high-density lipoprotein cholesterol (HDL-C) (**A**), low-density lipoprotein cholesterol (LDL-C) (**B**), ApoA-1 (**C**), and ApoB (**D**) were measured using a biochemistry analyzer. Non-HDL cholesterol values were calculated with the following equation: TC value − HDL-C value, which is presented in (**E**). Mice were grouped as indicated in Figure 1. Values are presented as means ± SEM (*n* = 7, each group). * *p* < 0.05, ** *p* < 0.01, and *** *p* < 0.001 compared with RD mice. ^†^
*p* < 0.05, ^† †^
*p* < 0.01, and ^† † †^
*p* < 0.001 compared with HFD mice.

**Figure 4 nutrients-10-01846-f004:**
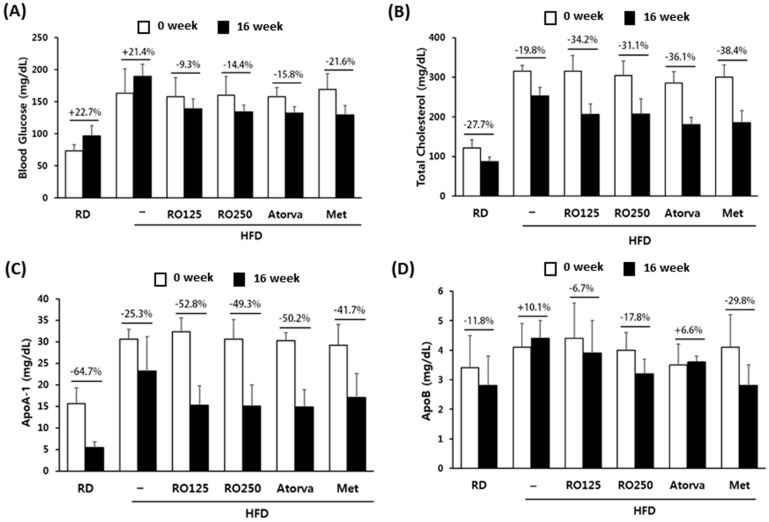
Representative biomarkers were compared before and after the oral administration of RO and drugs. Changes in blood glucose levels (**A**), total cholesterol (**B**), ApoA-1 (**C**), and ApoB (**D**) are presented with percentages. Mice were grouped as indicated in Figure 1. Values are presented as means ± SEM (*n* = 7, each group).

**Figure 5 nutrients-10-01846-f005:**
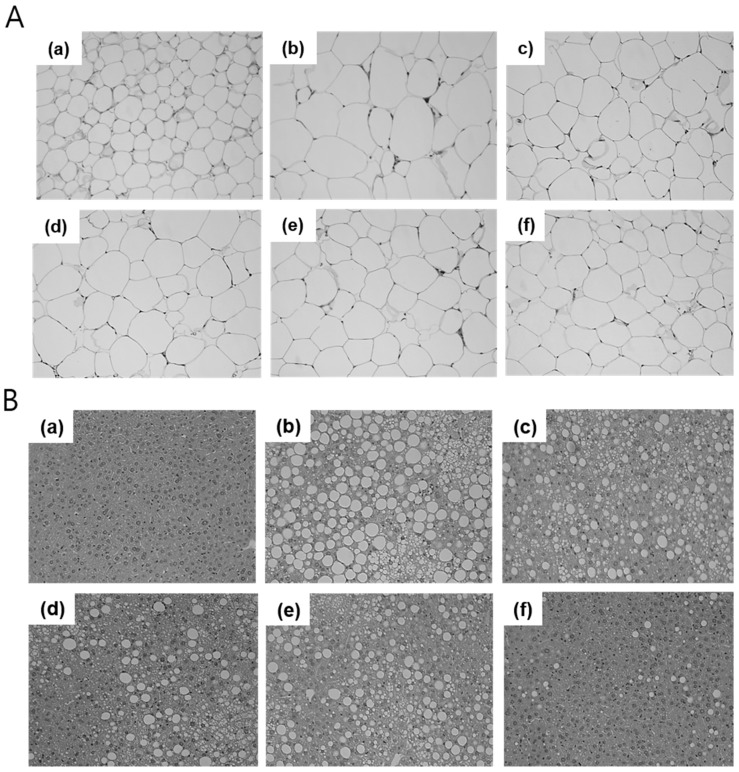
Representatives of each group were selected and their perirenal fat and liver tissue were stained with hematoxylin and eosin (H&E). Samples from the fat pad (**A**) and liver (**B**) were observed through a microscope at ×200 magnification (*n* = 1, each group). RD: regular diet (**a**), HFD: high-fat diet (**b**), HFD + RO 125: HFD-fed mice treated with low dose (125 mg/(kg·day)) of RO (**c**), HFD + RO 250: HFD-fed mice treated with high dose (250 mg/(kg·day)) of RO (**d**), HFD + Atorva: HFD-fed mice treated with atorvastatin (10 mg/(kg·day)) (**e**), HFD + Met: HFD-fed mice treated with metformin (250 mg/(kg·day)) (**f**).

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
