# Peer review of "Multifaceted Effect of Rubus Occidentalis on Hyperglycemia and Hypercholesterolemia in Mice with Diet-Induced Metabolic Diseases"

_nutrients, 2018, doi:10.3390/nu10121846_

Round 1

Reviewer 1 Report

Metabolic syndrome (MetS) is a plurimetabolic condition associated with obesity, insulin-resistance, and hyperlipidemia. MetS is a major risk factor for type 2 diabetes and cardiovascular disease. Diet is a key factor in the prevention of MetS and the related metabolic disorders.

Kim et al. investigated the impact of extract of Rubus occidentalis (RO), a berry with high content in antioxidants, on adiposity, insulin-resistance and lipid metabolism in mice with diet-induced metabolic disease.  

Briefly, investigators treated mice with diet-induced metabolic disease with either a low-fat diet, a high-fat diet, a high-fat diet with RO (2 different doses), a high-fat diet with statin, and a high-fat diet with metformin. Authors observed that RO extract induced reductions in blood glucose, Total-C, ApoA-1 and ApoB.

Authors concluded that RO simultaneously exerted antihyperglycemic and antihyperlipidemic effects in mice with diet-induced metabolic disease.

This is an interesting study. However, there are several issues related to the rationale of the study, the methods as well as the interpretation of the results, that need to be addressed by the authors.

-        Authors refer to “metabolic disease” as a condition highly similar to metabolic syndrome. It is however not clear if authors refer to metabolic syndrome per se. If so, “metabolic disease” should be replaced by “metabolic syndrome” throughout the paper.

Abstract:

-        Line 20: It is highly arguable that Rubus occidentalis has “well-known” antioxidant, anti-cancer and anti-inflammatory properties. Please rephrase.

-        Authors are invited to provide more detailed results than only P values in the abstract.

-        Line 25: Please specify that decreases were observed in apoA-1 and apoB.

Introduction:

-        Authors are invited to provide a more detail description of what RO is.

-        Line 37: the role of a “fatty diet” in the development of the metabolic syndrome is arguable. The use of “poor quality diet” would be more correct. Please rephrase.

-        Line 38: Authors provide a vague and unclear definition of the metabolic syndrome, although it has a very clear definition. Please rephrase.

Methods:

-        Authors are invited to clarify why they used statin and metformin as control treatments. It is also unclear what the use of statin and metformin add to study originality.

-        Did authors assess the amount of food consumed by mice in each treatment? It is currently not specified in the manuscript. One can argue that mice with RO, statin or metformin ate less, that could explain the results. Please clarify.

Results:

-        Figures 1A and 1C are too small. It is currently impossible to evaluate if interpretation of results is correct.

-        Figure 4 presents pre-post data on blood glucose, Total-C, ApoA-1 and ApoB. On the figure, we can see that pre-levels were not equal in each group. Did authors adjust for pre-levels in their analysis?

-        In my opinion, figure 4 is the most informative as it presents pre- and post levels. Did authors compare the treatment-induced reduction in each group? That would provide a more comprehensive assessment of the effect of each treatment, especially since pre-levels were not equivalent in each group.

Discussion:

-        Authors are invited to specify to what quantity for humans the dose of RO they used refers to.

-        Line 241-242: CETP does not “transform” HDL to LDL or VLDL. It promotes the exchange of cholesteryl-ester and TG between these lipoproteins. Please rephrase.

-        Line 265: It is unclear whether authors refer to HDLs and LDLs (the lipoproteins) or HDL-C and LDL-C (what they measured in the study). Please clarify.

Author Response

Hello.

I'm Hyunseok Kong.

Thank you for your comments about our research.

-        Figure 4 presents pre-post data on blood glucose, Total-C, ApoA-1 and ApoB. On the figure, we can see that pre-levels were not equal in each group. Did authors adjust for pre-levels in their analysis?

-        In my opinion, figure 4 is the most informative as it presents pre- and post levels. Did authors compare the treatment-induced reduction in each group? That would provide a more comprehensive assessment of the effect of each treatment, especially since pre-levels were not equivalent in each group.

-> We have selected the DIMD mice and  the selected mice was selected again in each group, but the pre-leves were not equal due to some standard level in the process. Therefore, the pre-and post levels of each group were checked and compared.

* The remaining comments were attached to the file.

Thank you

Reviewer 2 Report

The study titled, “Multifaceted effect of Rubus occidentalis on hyperglycemia and hypercholesterolemia in mice with diet-induced metabolic disease,” investigated whether blackberries improved blood glucose and cholesterol levels in mice with diet-induced metabolic syndrome. 

The research topic is interesting and fairly well written. 

I have the following comments for the authors to improve the manuscript.

1.    Rather than saying, Rubis occidentalis was extracted following standard method and chemically analyzed as in a previous study, it would better to briefly describe the methods to improve understanding.

2.    What was the purpose of dissolving the RO in carboxymethylcellulose? 

3.    Line 82: why was a >150 mg/dL used to define blood glucose? In humans, blood glucose level of ≥110 mg/dL is used metabolic syndrome and ≥126 is used to define diabetes. In addition, the authors said that animals with blood glucose >150 mg/dL were selected for lipid profiling. Why wasn’t a cholesterol cutoff used instead of glucose cutoff?

4.      Line 86: What are the average values listed 161.6 mg/dL, 304 mg/dL…? 

5.    Line 132: Were the control mice treated with atorvastatin or metformin or both? If only one, why both drugs were not given?

6.    Line 126-27: When was the stated starting time, a time before feeding started or before treatment started?

7.    It is difficult to distinguish lines in Figure 1 a and c. Please use different line types and/or colors to distinguish them.

8.    In the discussion, Line 205-: The first paragraph has to be about the major finding in your study. You may begin from the second paragraph (Line 215) and describe the main finding of your study. In the next paragraph compare your result with previous studies.

9.    Line 230: You state that several studies reported that there is no difference in TG between RD and HFD mice groups but cited only two studies. It would be better to cite those studies.

10.    Throughout the discussion, please do not refer back to the tables or graphs. 

11.    Line 237: Which pathogenesis are you referring to?

12.    Line 235-252: If mice do not have cholesteryl ester transport protein, what is the mechanism through which blackberries (Rubus occidentalis) reduced HDL?

13.    Please add, the strength, limitation, and implication of the study.

Author Response

Hello.

I'm Hyunseok Kong.

Thank you for your comments about our research.

-why was a >150 mg/dL used to define blood glucose?

-> We were attached to reference

* The remaining comments were attached to the file.

Thank you

Reviewer 3 Report

Kim et al have treated HFD-fed mice with Rubus occidentalis (which apparently has antioxidant effects) in an intervention setup and have observed improved glycemia, cholesterolemia, hepatosteatosis and lipid accumulation in fat tissue. I especially like the setup (intervention and not prevention), the rather long duration of pre-HFD (23 weeks) and intervention (16 weeks) and the matching of the mice pre-treatment to get the “DIMD”model as the authors call it. However, there are a few important points to be clarified prior to moving on with this article.

Major

·       Methods line 91-94: Were the control groups administered a vehicle control? If so, please add it to methods. Control groups need to be treated as well, especially as drug administration was very frequent (daily for 16 weeks).

·       Line 207. Novelty. The authors state that “…no animal models that reflect the practical traits of metabolic disease have been introduced to analyze the complex efficacy of RO”. But cited references 7 and 9 have exactly done that. The show very similar results as the current study. Therefore, the current study seems not to be that “novel”.

·       Body weight gain. References 7 and 9 show a very strong effect of RO on body weight gain. Why did the authors not see this in the current study? Is this because the current study has an intervention protocol rather than a prevention? But this should be discussed and more credit should be given to references 7 and 9.

·       Intervention vs prevention: the current study has an intervention setup rather than a prevention as the other cited studies. This is great, as in a clinical setting we are also dealing with intervention rather than prevention. However, it is more difficult to see effects in interventions studies. Nevertheless, this should be highlighted as the intervention setup is a real advantage of this current study.

Minor

·       Methods line 79: Please add the source of fat (coconut oil, lard, …) as this can heavily influence the degree of impairment

·       Methods line 84-86: Please add whether blood glucose was measured in a fed or fasted (fasted for how long, what time of day) state

·       Methods line 86-88: Please add deviations and/or minimum/maximum values to get an idea of the range

·       Results line 127: There seems to be a language problem here: “Reduced to 25%” means that something is reduced by 75% and the value is 25% of the initial value (100%). What the authors probably mean is “reduced by 25%” (which means that the end value is 75% of the initial value at 100%).

·       Results: symbol size in graph. Can the symbol sizes be increased? It is sometimes impossible to discern the lines from each other as some symbols look the same (e.g. Fig. 1C)

·       Figure legends: please add the number of mice that were used (“n”). Sometimes it is stated, but sometimes not

·       DOI Ref. 7 is wrong

Author Response

Hello.

I'm Hyunseok Kong.

Thank you for your comments about our research.

·       Line 207. Novelty. The authors state that “…no animal models that reflect the practical traits of metabolic disease have been introduced to analyze the complex efficacy of RO”. But cited references 7 and 9 have exactly done that. The show very similar results as the current study. Therefore, the current study seems not to be that “novel”.

·       Body weight gain. References 7 and 9 show a very strong effect of RO on body weight gain. Why did the authors not see this in the current study? Is this because the current study has an intervention protocol rather than a prevention? But this should be discussed and more credit should be given to references 7 and 9.

-> Cited references 7 and 9 were researched about Rubus coreanus.But our article was researches about Rubus occidentalis. So we think there is a difference in materials.

* The remaining comments were attached to the file.

Thank you

Round 2

Reviewer 1 Report

Changes made to the manuscript improved its quality. On the other hand, some comments were not adequately addressed. For instance, it remains unclear why investigators treated mice with statin and metformin. The rationale behind this methodological choice is not clearly presented. Moreover, it is felt that authors reply to my comments on figure 4 does not adequately address the point I raised about the unequal baseline levels.

Author Response

Hello.

I'm Hyunseok Kong.

Thank you for your comments about our research.

<First Review>

1. “metabolic disease” was replaced by ‘’metabolic syndrome” throughout the paper.

2. Abstract: P values was removed.

3. Figures 1A and 1C was resized.

4. HDLs and LDLs was replaced by HDL-C and LDL-C throughout the paper.

5. The tables or graphs was not refer in discussion.

6. Other changes to manuscript were highlighted by red color

<Second Review>

1. Comment: It remains unclear why investigators treated mice with statin and metformin. The rationale behind this methodological choice is not clearly presented.

Answer: Atorvastatin is a "high-intensity statin", an HMG CO-A reductase inhibitor that reduces LDL-C by more than 50%, and is currently one of the most commonly used drugs in clinical practice [20]. And, Metformin is an antidiabetic agent that has been clinically used for decades in the treatment of type 2 diabetes mellitus and has proven effective in improving [21]. For this reason, mice were treated with hypercholesterolemia and hyperglycemia, respectively, as control drugs

Changes to manuscript were highlighted by yellow lined

2. Comment: It is felt that authors reply to my comments on figure 4 does not adequately address the point I raised about the unequal baseline levels.

Answer: The table below shows the data for pre-levels. Average pre-levels of biomarkers except ApoB was significantly higher in HFD-fed mice than in RD fed mice. But, there were no significant differences between DIMD mice (HFD). Therefore, we think there is no problem in comparing the treatment effect because there is not a significant difference between the HFD groups although there is a slight difference in the pre-levels.

Group

HFD

RD

HFD

RO   125

RO 250

Atorva

Met

Blood   Glucose (mg/dL)

73.0±9.6

163.4±37.7 †††

157.9±29.7

159.9±29.3

157.6±14.5

169.4±23.9

Total Cholesterol (mg/dL)

120.9±21.5

315.1±15.4 ††

314.7±40.4

304.1±36.9

285.1±28.6

300.7±30.8

ApoA-1   (mg/dL)

15.7±3.6

30.7±2.2  

32.4±3.2

30.7±4.5

30.3±1.9

29.2±4.9

ApoB (mg/dL)

3.4±1.1

4.1±0.8

4.4±1.2

4.0±0.6

3.5±0.7

4.1±1.1

3. Comment: I still think that it would help visibility if the symbols in the graphs were bigger (the other reviewer actually had the same comment), but in the end it is up to the journal editor I guess. 

Answer: We checked the graphs once again and modified it as advised. Please check again.

4. Comment: For a future next manuscript that you will submit, it would be nice as a reviewer to get additionally a manuscript in which changes are highlighted (e.g.  by color, or underlined). If it is just a revised manuscript it takes a lot of time and energy to find all the things that changed. Also, it would be useful to get a point-to-point answer for all questions asked.

Answer: Sorry for not checking this part in detail. We highlighted the changes and attached the files. Please check again. Thank you for your advices.

Reviewer 3 Report

Kim et al have answered most questions and added the requested information. I still think that it would help visibility if the symbols in the graphs were bigger (the other reviewer actually had the same comment), but in the end it is up to the journal editor I guess. 

For a future next manuscript that you will submit, it would be nice as a reviewer to get additionally a manuscript in which changes are highlighted (e.g.  by color, or underlined). If it is just a revised manuscript it takes a lot of time and energy to find all the things that changed. Also, it would be useful to get a point-to-point answer for all questions asked. 

Author Response

(The authors gave the same response as above.)
